# Absence of Light Exposure Increases Pathogenicity of *Pseudomonas aeruginosa* Pneumonia-Associated Clinical Isolates

**DOI:** 10.3390/biology10090837

**Published:** 2021-08-27

**Authors:** Cristina S. Mesquita, Artur Ribeiro, Andreia C. Gomes, Pedro M. Santos

**Affiliations:** 1CBMA—Centre of Molecular and Environmental Biology, Department of Biology, Campus of Gualtar, University of Minho, 4710-057 Braga, Portugal; cristina.mesquita@i3s.up.pt (C.S.M.); agomes@bio.uminho.pt (A.C.G.); 2CEB—Centre of Biological Engineering, Campus of Gualtar, University of Minho, 4710-057 Braga, Portugal; arturibeiro@ceb.uminho.pt

**Keywords:** cytotoxicity, host–pathogen interaction, light exposure, pathogenicity, photobiology, *Pseudomonas aeruginosa*, virulence factors

## Abstract

**Simple Summary:**

An often-overlooked factor when designing an experiment is light exposure. Bacterial cultures are usually grown in an uncontrolled light environment, subject to natural daylight conditions. However, light as an abiotic factor has been described as a trigger for the switch from environmental to a pathogenic host-associated lifestyle. *Pseudomonas aeruginosa* possesses features that allow it to sense light and adapt its behavior accordingly. In this study, clinical isolates of *P. aeruginosa* from the sputum of pneumonia-diagnosed patients were assayed in two extreme light exposure conditions: constant illumination and intensity of full-spectrum light, and complete absence of light. The aim was to understand the influence of this factor on *P. aeruginosa* pathogenic potential, studied in interface with a human pulmonary epithelial cell line model. The preliminary findings here described evidenced that growth in the dark, which would more closely mimic the conditions in the lung microenvironment, was associated with a higher cytotoxicity against host cells, concordant with an increased pathogenic potential. Given the importance of accurately interpreting bacterial responses in in vitro studies, this study raises awareness of the need to control light exposure conditions to be as similar as possible to in vivo parameters.

**Abstract:**

*Pseudomonas aeruginosa* can alter its lifestyle in response to changes in environmental conditions. The switch to a pathogenic host-associated lifestyle can be triggered by the luminosity settings, resorting to at least one photoreceptor which senses light and regulates cellular processes. This study aimed to address how light exposure affects the dynamic and adaptability of two *P. aeruginosa* pneumonia-associated isolates, HB13 and HB15. A phenotypic characterization of two opposing growth conditions, constant illumination and intensity of full-spectrum light and total absence of light, was performed. Given the nature of *P. aeruginosa* pathogenicity, distinct fractions were characterized, and its inherent pathogenic potential screened by comparing induced morphological alterations and cytotoxicity against human pulmonary epithelial cells (A549 cell line). Growth in the dark promoted some virulence-associated traits (e.g., pigment production, LasA proteolytic activity), which, together with higher cytotoxicity of secreted fractions, supported an increased pathogenic potential in conditions that better mimic the lung microenvironment of *P. aeruginosa*. These preliminary findings evidenced that light exposure settings may influence the *P. aeruginosa* pathogenic potential, likely owing to differential production of virulence factors. Thus, this study raised awareness towards the importance in controlling light conditions during bacterial pathogenicity evaluation approaches, to more accurately interpret bacterial responses.

## 1. Introduction

*Pseudomonas aeruginosa* is an opportunistic pathogen frequently reported as the causative agent of healthcare-associated infections, particularly as the second most common cause of nosocomial pneumonia and the most common multidrug-resistant Gram-negative pathogen causing pneumonia in hospitalized patients [1]. The outcome of a *P. aeruginosa* infection depends both on the host response and the bacterial virulence factors that counter it [2,3]. Its ability to exert cytotoxicity, largely mediated by biofilm formation and its arsenal of secreted virulence factors [3], differs from isolate to isolate, growth conditions, and distinct stages of infection [4].

*P. aeruginosa* biofilms are encased by an exopolysaccharide matrix that protects cells from the host immune response, hindering eradication [5]. A well-known virulence factor is the LasA elastase, a protease associated with *P. aeruginosa* virulence in clinical situations by enhancing its ability to invade tissues and interfering with host defense mechanisms [6]. The secretion of pyocyanin, a blue redox-active phenazine, has been positively associated with cytotoxicity against host cells, as pyocyanin-deficient strains, which produce reduced levels or no pyocyanin, were associated with lesser cytotoxicity in a variety of hosts [7]. Pyoverdine is a siderophore, which also proved to be important for *P. aeruginosa* virulence, functioning as a toxin by removing iron from mitochondria, inflicting damage on this organelle [8].

Additionally, *P. aeruginosa* naturally releases outer-membrane vesicles (OMVs), spherical and bilayered structures that act as key players in bacterial communication and in the host–pathogen interaction. Owing to the nature of its cargo, the release of OMVs is considered a bacterial protective mechanism to endure aggressive host environment and its influence on bacterial pathogenesis has been reported [9]. In Gram-negative, these vesicles operate as a separate secretory system, allowing the delivery of proteins that increase their invasive abilities, virulence factors, toxins, and other immunomodulatory compounds, thus enhancing bacterial survival inside the host. Therefore, fractioning of the bacterial culture and its supernatant portion is required to identify and study the effect of distinct stresses on these virulence determinants.

*P. aeruginosa* is greatly associated with molecular flexibility and adaptability, being able to respond to distinct stresses by active modulation of antibiotic resistance, metabolic and secretory-related systems, among others [10]. Its pathogenicity is influenced by environmental factors, light being one of the known triggers for the switch from an environmental to a pathogenic host-associated lifestyle [11]. Resorting to photoreceptors, as the bacteriophytochrome BphP already described in *P. aeruginosa* [12], bacteria can sense different light wavelengths and regulate cellular processes ranging from energy production to virulence [13].

Recent studies have shown the ability of light to repress biofilm formation in *P. aeruginosa* reference strain PA14 [14,15], thus, we aimed to address how exposure to light as an abiotic factor affects the dynamic and adaptability of the reference strain PAO1 [16] and two *P. aeruginosa* pneumonia-associated isolates, HB13 and HB15 [17]. In silico analysis indicated the presence of the bacteriophytochrome BphP, which regulates light responses via reversible photoconversion between red and far-red light-absorption states, and conservation of its domains [12] in the genome of these *P. aeruginosa* isolates, suggesting the ability to sense light (Appendix A). The bacteriophytochrome BphP (PA13_1006205, PA15_0309150, PA4117) assembles with biliverdin, its chromophore, produced by the heme oxygenase BphO (PA13_1006200, PA15_0309145, PA4116) to generate a photosensing domain that is activated by light. A previous study has reported that, in *P. aeruginosa*, light-sensing stimulates autophosphorylation of residue H513 in the histidine kinase domain of BphP and subsequent transfer of the phosphoryl group to residue D59 of AlgB (PA13_1026260, PA15_0320210, PA5483) to activate it [14]. These residues were conserved in the three *P. aeruginosa* screened in this study, and BphP and AlgB possessed no putatively deleterious variation, thus implying no disparity in the ability to sense light. The output domain of this dimeric bacteriophytochrome signals to a downstream response regulator of a two-component regulatory system, being the target for such regulation often gene expression. Understanding the influence of light on the host–pathogen interaction, currently understudied, is essential for the design of new strategies to understand its pathogenicity.

In this preliminary work, two opposing growth conditions (constant presence and intensity of full-spectrum light and total absence of light) were characterized phenotypically. Furthermore, given the multifactorial and combinatorial nature of *P. aeruginosa* pathogenicity [4], and as light might influence the host–pathogen interaction by modulating its pathogenicity, distinct fractions were screened against a human cell model. As the two clinical isolates were collected from the sputum of pneumonia-diagnosed patients, the A549 human pulmonary epithelial cell line model [18] was selected.

## 2. Materials and Methods

### 2.1. P. aeruginosa Isolates

Two clinical isolates of *P. aeruginosa*, HB13 and HB15, and the reference strain PAO1 were used in this study. The previously characterized isolates [17] were obtained from sputum samples of clinically diagnosed pneumonia patients attending a Portuguese Hospital (Hospital de Braga). All bacterial isolates were maintained in *Pseudomonas* isolation agar (PIA, BD Difco™) plates for a maximum of 3 passages and, for liquid cultures, inoculated with a starting OD_600_ of 0.08 in 50 mL of Lysogenic Broth (LB; 1.0% (*w*/*v*) tryptone 0.5% (*w*/*v*) yeast extract and 0.5% (*w*/*v*) NaCl) in 250 mL flasks for 24 h at 37 °C with orbital shaking of 200 rpm. Resorting to a light insulated Panasonic MIR-154-PE Cooled Incubator, two opposing growth conditions were evaluated: (i) exposure to full-spectrum light at 4000 lumens to mimic sunlight exposure conditions (Light); and (ii) total absence of light (Dark).

### 2.2. Phenotypic Profiling of P. aeruginosa

Bacterial isolates were phenotypically categorized under the two opposing growth conditions (Light vs. Dark). Antibiotic susceptibility profiling followed the Clinical and Laboratory Standards Institute guidelines [19]. The determination of the minimum inhibitory concentration was conducted using the broth microdilution method [20] for six antipseudomonal agents (the aminoglycosides tobramycin and gentamicin, the carbapenem meropenem, the third-generation cephalosporin ceftazidime, the fluoroquinolone ciprofloxacin, and the polymyxin colistin). Biofilm formation was assessed according to Zegans et al. (2009) [21]. The production of two pigments was evaluated; specifically, pyocyanin, as previously reported by Salunkhe et al. (2005) [22], and pyoverdine, according to Ackerley et al. (2004) [23]. Levels of LasA protease in *P. aeruginosa* supernatant fractions were determined by assessing the lysis of boiled *S. aureus* cells, leading to a decrease in the absorbance at 600 nm, according to Salunkhe et al. (2005) [22].

### 2.3. Preparation of the P. aeruginosa Fractions

Four fractions were collected from *P. aeruginosa* for characterization and cytotoxic assays (Appendix A). Bacterial cultures were harvested by centrifugation at 10,000× *g* at 4 °C for 10 min. The cell pellet was washed twice with cold 10 mM TrisHCl pH 7.4 plus 250 mM sucrose. Afterwards, it was resuspended in 10 mM TrisHCl pH 7.4 and sonicated (15 min; cycles of 2 sec burst ON with 9 sec burst OFF) on ice. After sonication, the cell lysate was centrifuged at 5000× *g* at 4 °C for 15 min and the supernatant was filter sterilized through a 0.2 μm cellulose acetate syringe filter, resulting in the cytosolic fraction F1. The bacterial culture supernatant was centrifuged a second time at 10,000× *g* at 4 °C for 10 min and filtered through a 0.2 µm cellulose acetate syringe filter to remove any remaining cell debris, resulting in the secreted fraction F2. The secreted fraction was further separated by high-speed centrifugation at 50,000× *g* at 4 °C for 1.5 h. The collected supernatant corresponded to the soluble portion F3, whereas the pellet was washed and resuspended in 10 mM TrisHCl pH 7.4, resulting in fraction F4, supposedly comprising OMVs. The total protein content of all fractions was quantified by the Modified Lowry’s Method [24]. Aiming to assay the full potential of the *P. aeruginosa* fractions, and as freeze–thaw leads to a decrease in particle stability, even in OMVs [25], each fraction was used fresh after collection and directly administrated to the A549 cells.

### 2.4. Cell-Culture Maintenance

The A549 human adenocarcinoma epithelial cell line (ATCC CCL-185) was used to assess the pathogenic potential of *P. aeruginosa*. The A549 cells were maintained in high glucose Dulbecco’s modified Eagle’s medium (DMEM, from Sigma-Aldrich, St. Louis, MO, USA) F12 supplemented with 10% heat-inactivated fetal bovine serum (Biochrom, Cambridge, UK) and 1% L-glutamine with antibiotics (1% antibiotic mix-penicillin, streptomycin and amphotericin, from Sigma-Aldrich). Cells were grown at 37 °C in a humidified atmosphere with 5% CO_2_ and seeded every three days when confluence neared 90%.

### 2.5. Characterization of the P. aeruginosa Fractions

#### 2.5.1. DLS and Zeta Potential Analysis

To evaluate sample homogeneity and possible aggregation, the size distribution by intensity, the average particle diameter (Z-ave, d.nm), the polydispersity index (PDI, degree of non-uniformity of its size distribution), and the zeta potential (ZP, mV, surface charge) were measured at 25 °C using a Malvern Nano ZS Zetasizer. Purified bacteria fractions were diluted in phosphate buffer before analysis to match the highest concentration applied to the A549 cell lines. For both size and zeta potential measurements, a clear disposable zeta cell was used.

#### 2.5.2. Cell Morphology

Cellular morphology alterations in A549 cells induced by the distinct *P. aeruginosa* fractions were investigated by optical microscopy using an inverted microscope Olympus IX71 58F-3, equipped with an Olympus DP72 camera. The A549 cells were visualized directly in the 96 well plates destined for the cytotoxicity assay. ImageJ was used for the cell area determination, for comparison purposes, following the automated method described by Baviskar (2011) [26]. For this analysis, only cells seemingly viable were considered, without evidence of apoptosis hallmarks such as blistering and blebbing or formation of apoptotic bodies.

#### 2.5.3. Cytotoxicity Evaluation by the MTT Viability Assay

The viability of the A549 cells was determined by a colorimetric assay, MTT (3-(4,5-dimethylthiazol-2-yl)-2,5-diphenyltetrazolium bromide), which assesses metabolic activity and, by extrapolation, cellular viability [27]. Briefly, cells were seeded at a density of 10,000 cells/well on 96-wells tissue culture plates at 37 °C in a 5% CO_2_ atmosphere. After 24 h of growth, the medium was refreshed and the cells were exposed to three different concentrations of each fraction from each clinical isolate, based on a previous determination of the half-maximal inhibitory concentration (IC50) for the *P. aeruginosa* whole supernatant fraction: IC50, IC50/8, and IC50/64. Three controls were added: (i) A549 cells in fresh medium—life control; (ii) A549 cells incubated with 30% DMSO—death control; and (iii) A549 cells incubated with LB—bacterial medium control.

After 24, 48, and 72 h of contact, cell metabolic activity was assessed using the MTT viability assay. The medium with the *P. aeruginosa* fractions was removed, fresh medium containing MTT solution was added to each well (1/10 V), and the mixture was incubated at 37 °C for 2 h. The MTT solution was carefully decanted, and the formazan crystals were dissolved in DMSO:ethanol (1:1 *v*/*v*) and spectrophotometrically measured at a wavelength of 570 nm in a microplate reader (SpectraMax Plus 384, Molecular Devices, San Jose, CA, USA). The percentage of cell viability was determined in relation to the life control (100%). Absolute IC50 was determined by fitting a sigmoid-dose response curve to the data from a serial range of concentrations, using nonlinear regression (Table 1). Additionally, the growth of the A549 cells was monitored from initial plating to the three MTT timepoints, by cell counting. As cells exposed for 72 h seemed to reach a plateau of cytotoxicity, further comparisons considered the cellular viability obtained for the 48 h endpoint.

### 2.6. Statistical Analysis

Data were presented as the mean value and standard deviation of three independent experiments. Each independent experiment included at least three technical replicates for phenotypic profiling and four replicates for the MTT assay. Statistical comparisons were performed by two-way ANOVA with Prism version 8.4 (GraphPad Software, San Diego, CA, USA). Tukey’s post hoc test was used for multiple comparisons between results, and a Dunnett’s test was used to compare the results with a control. A *p*-value < 0.05 was considered significant.

## 3. Results

### 3.1. Phenotypic Plasticity Driven by Light Exposure

Comparing the phenotypic response of the three *P. aeruginosa* grown in the two extreme luminosity conditions, a difference in the production of key pathogenesis mediators became apparent. Despite showing no differences in the antibiotic susceptibility profile, other virulence-associated phenotypes were altered in response to this abiotic factor (Figure 1 and Figure 2).

The assayed traits of biofilm and pigment production were positively affected when the clinical isolates were grown in the dark. The same trend was observed for LasA protease activity, even if only significant for *P. aeruginosa* HB15.

### 3.2. Assessment of the Pathogenic Potential of P. aeruginosa

Owing to the multifactorial and combinatorial nature of *P. aeruginosa* pathogenicity, distinct fractions of each isolate, when grown in opposing light regimen conditions, were characterized by DLS (Appendix A) and applied to cultured human pulmonary epithelial cells, to ultimately highlight relevant contributors to its pathogenicity.

The OMVs fraction (F4) seemed to be the one with a more distinctive profile and a more variable size distribution, suggesting a greater influence of the luminosity conditions in the OMVs profile of *P. aeruginosa*, along with a composition more dependent on the isolate. Additionally, despite not being significant for fractions F2 and F3, for *P. aeruginosa* isolate HB15, a trend was observed, with increased average particle size recorded in all fractions when the culture was grown in the dark.

Upon contact with the distinct fractions of *P. aeruginosa*, A549 cells, usually presenting a squamous shape and organized as an adherent monolayer, revealed morphological alterations, in a concentration- and time-dependent manner, and a correlation between cell density, average cell area, and cellular viability reduction (Figure 3, Appendix A).

Some cells presented features associated with a cytopathic effect, assuming a round shape due to changes in the actin structure and events of membrane blebbing and formation of vesicles, likely apoptotic bodies. Additionally, with increasing fraction concentration along with a reduction in cell density, A549 cells became increasingly smaller and more elongated, acquiring a spindle-shaped morphology, characteristic of the mesenchymal cell state.

Direct microscopy observation suggested a common ability of *P. aeruginosa* supernatant fractions to induce morphological alterations in the epithelial cells, possibly associated with the acquisition of a mesenchymal-like state. Nevertheless, the observed reduction in cell area was more evident in the A549 cells exposed to fractions collected after *P. aeruginosa* was grown in the dark, albeit without statistical significance (Figure 3b).

Regardless of the previous demonstration that the three *P. aeruginosa* strains exerted an equal effect on the viability of A549 cells, distinct cellular responses were observed for each fraction and growth condition (Figure 3b, Table 2).

Excluding the cytosolic fraction (F1) of the reference strain PAO1 and the clinical isolate HB13, the cytotoxic effect in A549 cells was always significantly higher (*p* < 0.001) when the *P. aeruginosa* cultures were grown in the dark.

## 4. Discussion

The pathogenic potential of *P. aeruginosa* is influenced by the production and secretion of virulence factors [4]. The observed phenotypic alterations in Figure 1 and Figure 2 suggest that the pathogenic potential of the tested *P. aeruginosa* isolates is affected according to the light exposure growth conditions, particularly its biofilm and pigment production. Recently, in agreement with our results, exposure of *P. aeruginosa* to light has been reported to repress biofilm formation [14].

To deliver a solid description of the pathogenic performance of each *P. aeruginosa* clinical isolate, the cytosolic and secreted portion, and further fractioning of the latter to isolate OMVs, were assayed against the human cell line A549 (Appendix A).

The observed cytopathic effect of cell rounding (Figure 3a) had been reported [28] and this cytoskeleton rearrangement, due to changes in the actin structure, may be caused by *P. aeruginosa* virulence factors [29]. Events of membrane blebbing and formation of apoptotic bodies are apoptosis hallmarks and could indicate that A549 cells are undergoing apoptosis under influence of *P. aeruginosa*, as previously reported [30]. Possible evidence of cell shrinkage, with the decrease in the A549 cells average area, an effect more evident for fractions collected from bacterial growths in the dark, could be another indicator of cells undergoing apoptosis. Furthermore, the observed acquisition of a spindle-shaped morphology by A549 cells, possibly associated with epithelial-to-mesenchymal transition, is a biological process where epithelial cells trade their polarity and cell–cell adhesion status, for improved migratory and invasive capacity [31]. Supporting these observations, the ability of *P. aeruginosa* to induce this process has been described [32]. Nevertheless, to confirm that this transition occurred within the cell model, molecular experiments, such as immunofluorescent cell staining and Western blotting, should be performed.

Sustaining former studies and in silico predictions of a higher virulence ability [17], the clinical isolate HB15 outperformed all the others, and a lower concentration of its fractions was required to exert a reduction effect on the viability of A549 cells (Table 1). As all secreted fractions (F2–F4) led to a reduction in cellular viability, it was suggested that the switch from light to dark resulted in a secretome content with a higher pathogenic potential, possibly associated with an increase in the production of virulence-associated traits, as deduced from the phenotypic characterization (Figure 1 and Figure 2) and hinted by previous identification of the secretome of these *P. aeruginosa* [17]. The cytotoxicity data obtained with pulmonary epithelial cells also corroborated a previous suggestion of association of isolate HB13 with a possible less cytotoxic [3], chronic persistent strain, whereas the traits evidenced by isolate HB15 are more common in an isolate from an initial colonization phase or an acute infection context.

Albeit being isolate-dependent, suggesting that the response to the light exposure conditions could be dynamically adapted by *P. aeruginosa*, this cytotoxic potential also proved to be fraction-dependent, and particularly from the secreted fractions, the OMVs fraction was the one with a more striking effect on A549 cell viability (Figure 3b). As key players in bacterial communication and in the host–pathogen interaction [9], the natural secretion of OMVs is here suggested to potentially stimulate a strain-dependent host response.

This study evidenced that, indeed, the light regimen influenced the pathogenic potential of *P. aeruginosa* isolates and suggested that this influence is likely caused by differential production of virulence factors. This influence could also be verified when comparing the effective response on the viability of A549 cells exposed to a concentration of the whole supernatant fraction (F2), corresponding to the determined IC50 for a bacterial growth in an uncontrolled light regimen, when *P. aeruginosa* was grown in two opposing light exposure conditions (Table 2). Considering the example of *P. aeruginosa* HB15, the same concentration of the supernatant was associated with a 50% cellular viability, when collected from an uncontrolled growth, yet resulted in a viability of A549 cells of 62% and 39%, when collected from, respectively, growth in light or in dark.

All assayed virulence-associated traits were positively affected when the culture was grown in the dark, which, together with higher cytotoxicity associated with secreted fractions collected from the same bacterial growth, sustains an increased pathogenic potential of *P. aeruginosa* when grown under a complete absence of light. Considering that the clinical isolates HB13 and HB15 were collected from sputum samples, growth in the dark more closely mimics the light regimen that these isolates were subjected to in their original in vivo microenvironment, when in the lung of pneumonia-diagnosed patients. In this situation, bacteria would be switching from sunlight exposure to light absence inside the host, and the induced adaptations would be to increase the ability to invade the host.

The discovery of a *P. aeruginosa* photoreceptor (Appendix A) suggests that changes in luminosity conditions, being in the extreme settings of constant exposure or total absence as studied here, or exposure to specific wavelengths as light is an electromagnetic spectrum, could alter the activity of the photoreceptor and potentially impact the bacterial behavior. As the influence of light on both the virulence of pathogens and the host defense response has been discussed, this study verified that this abiotic factor impacted the *P. aeruginosa* pathogenic potential, particularly for two pneumonia-associated isolates.

Perceiving the changes triggered by differential light exposure and the underlying molecular mechanisms may aid in the development of new biotechnological tools, such as a light-controlled expression system, which would possess a wide range of applications in several fields.

To assess possible changes in the molecular dynamics induced by luminosity conditions, a preliminary cytosolic proteome comparison has suggested that exposure to light is mostly associated with the ability to sustain oxidative stress. This apparent state of oxidative stress could be owed to the continuous exposure to full-spectrum light, as it has been reported that bacteria irradiated with blue (415–470 nm) and red (620–700 nm) light displayed increased ROS production levels [33]. Moreover, the expression profiles associated with growth in the dark hinted higher overall stress tolerance, stimulation of quorum-sensing and possibly higher secretion of virulence factors in the absence of light, given that the latter were under-expressed in the cytosolic proteome. Nevertheless, future studies of the proteome of all screened *P. aeruginosa* fractions are necessary, to identify relevant contributors to the possible dynamic response to light exposure. Furthermore, to validate that the luminosity conditions are being detected by the *P. aeruginosa* bacteriophytochrome and influencing the production of virulence factors, a *bphP* mutant should be tested in the same growth conditions.

## 5. Conclusions

In summary, the light exposure conditions affect the results obtained for virulence-associated traits and host cell cytotoxicity. Given the origin of the screened *P. aeruginosa* clinical isolates, it could be postulated that growth in the dark more closely mimics the light exposure conditions that these isolates sustained in the host. These preliminary findings hint that the observed overall responses of *P. aeruginosa* are influenced by how the growth conditions differ from the natural luminosity conditions the microorganisms are adapted to. Consequently, it is imperative to maintain the experimental settings similar to in vivo conditions, including the often-disregarded luminosity conditions, in order to more accurately interpret the observed bacterial responses.

## Figures and Tables

**Figure 1 biology-10-00837-f001:**
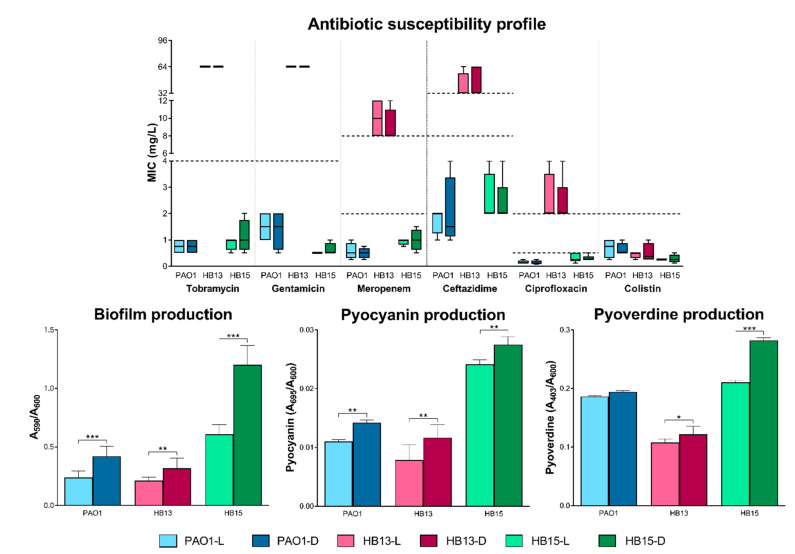
Phenotypic characterization of *Pseudomonas aeruginosa* grown in constant exposure to full-spectrum light (L) and total absence of light (D). Box and whiskers plots of the MICs of the six antibiotics screened, determined by broth microdilution. Horizontal dotted lines indicate the clinical susceptibility breakpoints [19]. Biofilm, pyocyanin and pyoverdine production quantified, respectively, by the absorbance measured at 590 nm [21], 695 nm [22], and at 403 nm [23], normalized by the cell density, monitored at 600 nm. Statistical comparisons performed by two-way ANOVA, followed by Tukey’s post hoc test for multiple comparisons. Significant differences comparing the two growth conditions are indicated as: * *p* < 0.05, ** *p* < 0.01, *** *p* < 0.001.

**Figure 2 biology-10-00837-f002:**
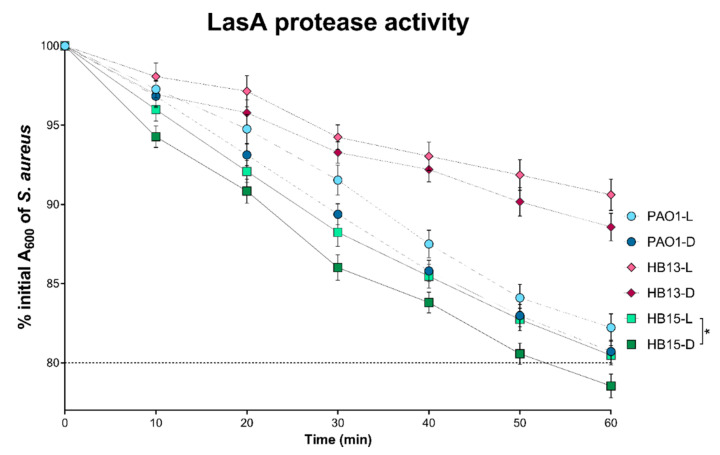
Levels of LasA protease activity of *Pseudomonas aeruginosa* grown in constant exposure to full-spectrum light (L) and total absence of light (D). LasA protease activity measured as the decrease in the absorbance at 600 nm of a *S. aureus* boiled culture [22]. The dotted line represents the cut-off for LasA overexpression, expressed as a final absorbance lesser than 80%. Statistical comparisons performed by two-way ANOVA, followed by Tukey’s post hoc test for multiple comparisons. Significant differences comparing the two growth conditions are indicated as: * *p* < 0.05.

**Figure 3 biology-10-00837-f003:**
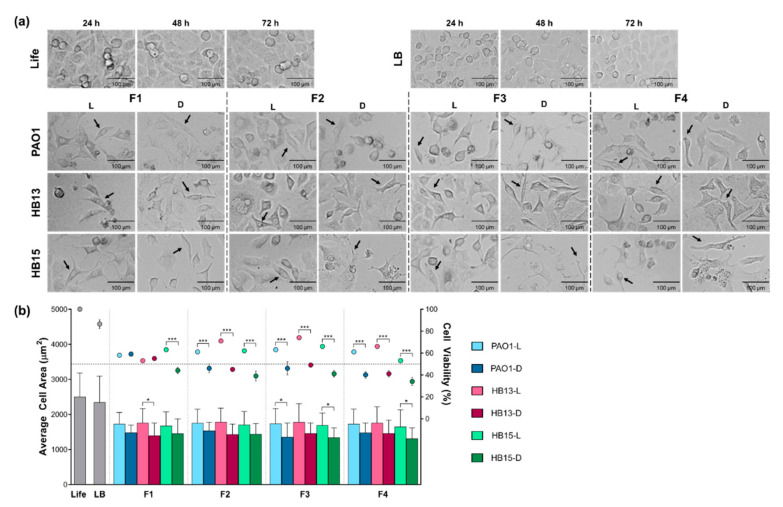
Effects induced in A549 cells by fractions of *P. aeruginosa* grown in constant exposure to full-spectrum light (L) and total absence of light (D). Four freshly collected fractions were screened: the cytosolic fraction (F1), the whole supernatant (F2), the further fractioned soluble portion (F3), and the OMVs fraction (F4). (**a**) Morphologic alterations induced in A549 cells by a specific fraction concentration (Table 1) at 48 h endpoint. A representative Life and LB control for the three MTT endpoints is presented. Arrows highlight few examples of cells representing a possible transition to a mesenchymal-like state. The scale bar is set for 100 µm. Full contrast phase microscope shot available in Appendix A. (**b**) Graphical representation of determined average area (bars) and viability (circles) of A549 cells. A horizontal line highlights a cellular viability of 50%. Standard deviation bars are represented. Statistical comparisons performed by two-way ANOVA, followed by Tukey’s post hoc test for multiple comparisons. Significant differences comparing the two growth conditions are indicated as: * *p* < 0.05, *** *p* < 0.001.

**Table 1 biology-10-00837-t001:** Determined values of the IC50 (μg/mL). The A549 cells were screened for cytotoxicity at three MTT endpoints (24, 48, and 72 h) with the whole supernatant of a bacterial culture. All assays were repeated independently three times and each independent MTT assay included at least four technical replicates. Standard deviations (SD) are represented.

MTT Endpoint	IC50 (μg/mL) ± SD		
	PAO1	HB13	HB15
24 h	4.91 ± 0.14	3.73 ± 0.11	3.08 ± 0.08
48 h	4.66 ± 0.21	3.33 ± 0.07	2.79 ± 0.05
72 h	3.61 ± 0.05	2.91 ± 0.05	1.59 ± 0.04

**Table 2 biology-10-00837-t002:** Summary of the effective response on the viability of A549 cells (%) induced by fractions of *P. aeruginosa* grown in constant exposure to full-spectrum light (L) and total absence of light (D). Cells were incubated with a specific concentration (Table 1). Standard deviations are represented (±SD). The percentage variation on the effective response on the viability considering a light to dark switch was determined (Δ_L→D_), also taking in consideration the administrated concentration of each fraction (Δ_L→D_/IC50).

*P. aeruginosa*	Fraction	Cell Viability (%) ± SD		Δ_L→D_	Δ_L→D_/IC50
		L	D		
PAO1	F1	58 ± 2	59 ± 3	+1	+0.2
	F2	61 ± 2	46 ± 4	−15	−3.2
	F3	63 ± 3	46 ± 7	−17	−3.6
	F4	61 ± 3	40 ± 5	−21	−4.5
HB13	F1	53 ± 3	55 ± 1	+2	+0.6
	F2	71 ± 6	45 ± 1	−26	−7.8
	F3	74 ± 4	49 ± 2	−25	−7.5
	F4	66 ± 2	41 ± 3	−25	−7.5
HB15	F1	63 ± 5	44 ± 4	−19	−6.8
	F2	62 ± 4	39 ± 5	−23	−8.2
	F3	66 ± 3	41 ± 3	−25	−9.0
	F4	53 ± 3	34 ± 4	−19	−6.8

## Data Availability

Data are contained within the article or Appendix A.

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
