# Peer review of "Absence of Light Exposure Increases Pathogenicity of Pseudomonas aeruginosa Pneumonia-Associated Clinical Isolates"

_biology, 2021, doi:10.3390/biology10090837_

Round 1
Reviewer 1 Report
The aim of the study was a very interesting issue of treating Pseudomonas aeruginosa strains with light and dark and checking its impact on the strains behaviour. I suppose a lot of effort went into the study. The chosen approach, novelty and several findings made on the basis of the obtained results are quite interesting, in my opinion.
However, my major concerns include:
- What do the “design experiments” and the whole last sentence of the Abstract refer to? It should be explained, especially that the same confusing issue refers to the last sentence of the Conclusions section.
- A small number of the strains included into the study might have influenced the credibility of the results. Maybe it is not of the highest importance for the approach adopted in the research. However, the observations would be more significant for more representative number of strains.
- Introduction should be rewritten/shorten to explain only the most important issues of the research.
- In turn, OMVs mentioned in the Introduction section should be explained more detailed.
- What are the pneumonia criteria you mentioned and how was it confirmed/investigated– it should be explained.
- Why this particular set of four antimicrobials was used for AST? None of them is a drug of choice for the treatment of P. aeruginosa infections.
- Why CLSI guidelines were applied for the AST interpretation? What about EUCAST?
- The description of the applied methodology, in my opinion, is really superficial. It should be explained more sufficiently since it is rather sophisticated for a particular reader.
- Lack of any molecular studies showing a potential of the studied strains to synthesize the mentioned virulence factors, e.g. proteolytic activity may also result from other enzymes presence.
- Streptomycin was used for the cell culture purpose for well-known reasons. However, aminoglycosides have some influence on human cells proteins synthesis. Isn’t it possible for this antimicrobial to influence the growth of this particular cell line and the observed effect?
- Discussion section should be rewritten, it contains sentences that should be deleted or moved into the Introduction part, especially the first and second paragraphs.
- The conclusion that EMT process was observed looks like an exaggeration for me. No molecular experiment has been performed to confirm this observation.
My minor concerns are:
- The full name of bacteria should be used only once - first time mentioned in the text.
- In my opinion, some of the sentences from the Results section should be omitted or moved to Discussion part.
- The title should be changed into more general observation statement.
- Keywords should be listed in an alphabetical order, in my opinion.
However, all the points mentioned above do not reduce the substantive value of the research.
Author Response
The authors thank the reviewer1 for the comments. All the suggestions and comments of reviewer1 were considered in the revised version of the manuscript.
Regarding the comments made, please see the attachment.

Reviewer 2 Report
This preliminary report indicates an intriguing phenomenon of different pathogenic potential of Pseudomonas aeruginosa strains depending on presence/absence of light. This is an intersting study, despite no molecular mechanism was found.
Specific points:
- In fact, the authors discuss the BphP photoreceptor as a possible player in the regulation of pathogenicity in response to light conditions. It would be a straight-forward way to test this hypothesis by using a bphP mutant and test whether the light-dependence of the pathogenic potential is no longer oberved. However, this might be a subject for another study if the authors did not have results of such experiments ready.
- The authors may consider to divide Figure 1 for 3 separate figures. Panels with anibiotic susceptibility and LasA are very small and it is not easy to analyze them. Showing them as separate figures would enhance clarity of presentation.
- The authors are encouraged to present at least some of results included in the Supplementary material as regular figures. Again, this would enhance clarity of presentation.
Author Response
The authors thank the reviewer2 for the comments and suggestions, which were considered in the revised version of the manuscript.
Regarding the comments made, please see the attachment.

Round 2
Reviewer 1 Report
N/A